# Effect of Botulinum Toxin A on Bladder Pain—Molecular Evidence and Animal Studies

**DOI:** 10.3390/toxins12020098

**Published:** 2020-02-03

**Authors:** Ting-Chun Yeh, Po-Cheng Chen, Yann-Rong Su, Hann-Chorng Kuo

**Affiliations:** 1Division of Urology, Department of Surgery, Taiwan Adventist Hospital, Taipei City 105, Taiwan; breadtree100@gmail.com; 2Department of Urology, En Chu Kong Hospital, New Taipei City 237, Taiwan; b90401049@ntu.edu.tw; 3Department of Urology, National Taiwan University Hospital Hsin-Chu Branch, Hsinchu City 300, Taiwan; yrsu@hch.gov.tw; 4Department of Urology, Hualien Tzu Chi Hospital, Buddhist Tzu Chi Medical Foundation and Tzu Chi University, Hualien City 970, Taiwan

**Keywords:** botulinum toxin A, bladder pain, interstitial cystitis, molecular mechanism

## Abstract

Botulinum toxin A (BTX-A) is a powerful neurotoxin with long-lasting activity that blocks muscle contractions. In addition to effects on neuromuscular junctions, BTX-A also plays a role in sensory feedback loops, suggesting the potentiality for pain relief. Although the only approved indications for BTX-A in the bladder are neurogenic detrusor overactivity and refractory overactive bladder, BTX-A injections to treat bladder pain refractory to conventional therapies are also recommended. The mechanism of BTX-A activity in bladder pain is complex, with several hypotheses proposed in recent studies. Here we comprehensively reviewed properties of BTX-A in peripheral afferent and efferent nerves, the inhibition of nociceptive neurotransmitter release, the reduction of stretch-related visceral pain, and its anti-inflammatory effects on the bladder urothelium. Studies have also revealed possible effects of BTX-A in the human brain. However, further basic and clinical studies are warranted to provide solid evidence-based support in using BTX-A to treat bladder pain.

## 1. Introduction

Botulinum toxin, one of the most powerful neurotoxins in nature, is produced by the anaerobic, Gram-positive organism, *Clostridium botulinum*. Exposure to the botulinum toxin can be fatal, since this can lead to flaccid paralysis of the muscles, dysautonomia, and subsequent respiratory failure [1]. Of the seven distinct serotypes (A through G), botulinum toxin A (BTX-A) shows the longest duration of activity in blocking transmission at the neuromuscular junctions, making it the most popular form for clinical use. In 1988, Dykstra et al. were the first to use BTX-A in a urological application by injecting it into the urethral sphincter to treat detrusor sphincter dyssynergia in spinal cord injury patients [2].

Nowadays, BTX-A injection has been widely used in lower urinary tract diseases and is approved for patients with both overactive bladder (OAB) and neurogenic detrusor overactivity (NDO). In addition to OAB and NDO, using a BTX-A injection to treat the pain of interstitial cystitis/bladder pain syndrome (IC/BPS) is recommended in patients refractory to conventional therapies [3]. IC/BPS is a long-time challenge for urologists who treat its multifactorial conditions and accompanying pain. Recently, it was recognized that the disease not only has organ-specific syndromes, but also urogenital manifestations of regional or systemic abnormalities characterized by neuropathic pain [4].

The mechanism of BTX-A activity on bladder pain has been investigated: it possibly affects both afferent and efferent nerves, along with having an antinociceptive mode of action [5]. Here we reviewed current molecular and cellular evidence and related animal studies for a better general understanding of the mechanism of action of BTX-A in bladder pain.

## 2. Results

### 2.1. Basic Mechanism of Action of BTX-A

Inactive BTX-A is a single-chain polypeptide of 150 kDa. When BTX-A is pharmacologically activated, it is cleaved to a 100-kDa heavy chain and a 50-kDa light chain that are connected by a single disulfide bond as well as noncovalent bonds [1,6]. BTX-A inhibits or reduces muscle contractions by blocking vesicular neurotransmitter release at neuroglandular and neuromuscular junctions. Two types of presynaptic cell membrane surface receptors for BTX-A have been identified—gangliosides and the synaptic vesicle-associated protein-2 (SV2) family. BTX-A binds to nerve terminals because of the high affinity of its heavy chain for SV2 allowing the toxin to be endocytosed into synaptic vesicles [7]. The light chain of BTX-A is translocated across the vesicle membrane in an acidic environment, and is then released into the cytosol by reduction of the interchain disulfide bond. Following its release from vesicles, the light chain is able to cleave synaptosomal-associated protein 25 (SNAP25) proteins, a part of a heterotrimeric soluble N-ethylmaleimide-sensitive factor attachment protein receptor (SNARE) complex, thereby inhibiting the fusion of vesicles with the nerve terminal membrane, and ensuring the blockade of neurotransmitter release and consequent smooth muscle contractions [8].

When using BTX-A to treat lower urinary tract diseases, the net effect results in: (1) the paralysis of low-grade contractions of the unstable detrusor to increase bladder capacity and reduce detrusor pressure during filling and resting phases, and (2) the preservation of high-grade contractions of the detrusor to initiate micturition [9,10,11]. In addition to this effect, a significant reduction in the sensation of urinary urgency has been reported by patients with OAB, suggesting a sensory effect on the bladder [12]. The effects on sensory feedback loops explain the mechanism of BTX-A activity in relieving symptoms of detrusor overactivity as well as suggest a potential role for BTX-A in the relief of hyperalgesia-associated lower urinary tract disorders, such as IC/BPS and chronic pelvic pain syndrome [9].

### 2.2. BTX Effects on Peripheral Sensory Nerves

Chronic pain persisting from months to years can reduce people’s quality of life and become a major health-care burden. The pathophysiology of chronic pain can be classified as inflammation, neuropathic pain, or dysfunction [13]. Jankovic et al. described the first clinical application of BTX on cervical focal dystonia and hemifacial spasms. They observed an improvement in the pain of these patients, supporting an antinociceptive or afferent-mediated activity by BTX-A [14]. This finding initiated an era of research on the analgesic effects of BTX-A. Figure 1 shows the proposed mechanism of BTX-A effects on peripheral sensory system.

#### 2.2.1. Bladder Stretch (Spasm)-Related Visceral Pain

One possible pathomechanism of visceral pain has been proposed recently: tension-sensitive nerve terminals in the smooth muscle of hollow organs may respond to luminal distension or stretching [15,16]. In the bladder, BTX-A acts in detrusor muscle relaxation by inhibiting acetylcholine (Ach) release from parasympathetic nerve endings [17] (Figure 1, part B).

The transient receptor potential (TRP) superfamily of cationic ion channels is involved in many cellular functions and such channels are highly expressed in afferent neurons of the urinary bladder [18]. Members of the TRP channel superfamily include TRP vanilloid 1 (TRPV1), TRPV4, and TRP Ankyrin 1 (TRPA1), involved in the mechanosensory pathway of urothelial cells. Activation of such ion channels releases adenosine triphosphate (ATP), prostaglandin E2 (PGE2), and substance P, and causes visceral pain [18,19,20]. Therefore, it is inferred that the function of muscle paralysis by BTX-A is to help decrease bladder tension, reduce bladder spasms, downregulate such TRP channels, and consequently relieve bladder pain.

Chuang et al. had observed that intravesical BTX-A administration could significantly prolong the inter-contraction interval (ICI) of the bladder and produce analgesia against acetic acid-induced bladder pain in rats by inhibiting calcitonin gene-related peptide (CGRP) release from afferent nerve terminals [21].

When the bladder is distended, ligand-gated ion channel P2X purinoceptors 3 (P2X3) receptors on nerve endings in the bladder urothelium are activated by released ATP and evoke a neural discharge [22]. In an in vitro study, P2X3 subunits expressed by cultured IC bladder urothelial cells were upregulated during stretching; augmented ATP signaling in the bladder may explain IC symptoms [23]. Hanna-Mitchell et al. [24] and Collins et al. [25] demonstrated that the intravesical administration of BTX-A is effective in reducing stretch-induced ATP release in rats and mice models.

#### 2.2.2. Inhibition of Nociceptive Neurotransmitter Release in Peripheral Endings

The antinociceptive effects of BTX-A were initially considered in simple muscle relaxation. Recent studies found muscle relaxation effects may not directly overlap with pain relief, which implies that the mechanism of action of BTX-A in pain relief is more complex than first thought and may have possible effects on sensory neurons [26,27]. Of note, the analgesic effects of BTX-A often persist longer than the neuroparalytic effects, which indicates that BTX-A affects pain fibers or sensory nerves.

Unmyelinated C-fibers and lightly myelinated Aδ-fibers are two types of nerve fibers that transmit sensory information from the bladder [28,29]. C-fibers are the primary nociceptive fibers that innervate the suburothelium of the bladder. Under normal conditions, C-fibers are silent but become activated in several pathological conditions, such as the alteration of potassium channels. Activated C-fibers result in increased excitability, the transmission of painful stimuli, and increased afferent drive that consequently contributes to detrusor hyperreflexia [29].

Nerve growth factor (NGF) influences C-fiber hyperexcitability in studies done by Vizzard et al. [30] and Seki et al. [31]. They observed increased NGF levels in the bladder, spinal cord, and lumbosacral dorsal root ganglia (DRG) in animals with spinal cord injury or chronic cystitis that exhibited bladder overactivity. Yoshimura et al. [32] set up a study to mimic this mechanism by intrathecally injecting NGF into female rats. Cystometrograms showed a reduction in ICI and voided volume, indicating bladder overactivity. Liu and Kuo [33] confirmed that intravesical BTX-A treatment can reduce NGF production to a normal level and control pain in IC/BPS patients.

C-fibers also release neuropeptides such as CGRP and substance P (SP), which are upregulated in patients with IC/BPS [34]. Numerous studies have shown that BTX-A blocks the release of nociceptive neurotransmitters from peripheral sensory nerves [35]. Durham et al. provided the first evidence that BTX-A directly decreases the release of CGRP from trigeminal neurons [36]. Welch et al. reported that BTX-A inhibits the release of CGRP, glutamate, and SP from cultured embryonic rat DRG [37]. BTX-A inhibition of the stimulated release of CGRP and SP from afferent nerve terminals has also been confirmed ex vivo by Lucioni et al. in rats with cyclophosphamide (CYP)-induced cystitis [38] and Rapp et al. in a capsaicin-evoked rat bladder model [39]. The antinociceptive effects of BTX-A in peripheral endings are depicted in Figure 2.

TRPV1 is a vanilloid receptor expressed in C-fibers that are involved in pain transmission after activation by heat, capsaicin, or resiniferatoxin [40]. A previous study showed that increased severity of inflammation correlated with a higher expression of TRPV1-immunoreactive nerve fibers and NGF levels in bladder biopsies from IC/BPS patients [41].

In addition to C-fibers, BTX-A may also affect Aδ-fibers. In an in vitro mouse preparation, BTX-A weakened both low- and high-threshold afferent units firing during bladder distension, suggesting that it acts on both Aδ- and C-fibers [25]. This same study revealed a five-fold increase in nitric oxide (NO) in urothelium after BTX-A administration. The increase in NO may have diminished afferent activity by acting as an inhibitory transmitter on myofibroblasts or the urothelium [42]. A balance in urothelial neurotransmitters between excitatory ATP and inhibitory NO has been proposed to modulate afferent activity and it would seem that BTX-A contributed to normalizing this balance [43].

#### 2.2.3. Anti-inflammatory Effects of BTX-A in Bladder Urothelium

Inflammation-induced afferent sensitization to recruit immune cells is an important protective mechanism to resist infection in the urinary tract [44,45] (Figure 1, part C). However, prolonged inflammation may result in the long-lasting sensitization of afferents and lead to chronic pain [45]. Most experts believe that bladder pain can be attributed to chronic inflammation although the pathogenesis is still inconclusive due to contradictory histopathology results. However, an increase in proinflammatory mediators (histamine, NGF, and those released from mast cells) within the bladder and urine of IC/BPS patients has been widely reported, which is consistent with the hypothesis of inflammation causing bladder pain [46,47,48]. Cui et al. [49] were the first to describe the direct involvement of BTX-A in pain modulation after inhibiting inflammation. Formalin-induced edema and accompanying peripheral glutamate release were reduced by intraplantar BTX-A injection in rats.

To investigate the inflammatory-mediated pathophysiology of bladder pain, a range of irritants or immune stimulants, including CYP, lipopolysaccharide, acetic acid, acrolein, and protamine sulfate, have been administered in animal models [50]. Altered cystometry and an enhanced visceromotor response during bladder distension were shown in experimental animals, which mimicked a reduced bladder capacity and the hyperalgesia and allodynia observed in response to bladder distension in humans [45,51].

Chuang et al. [52] also reported that BTX-A intravesical injection reduced cyclooxygenase 2 (COX2) and PGE2 receptor expression in the bladders of rats with CYP-induced cystitis. In humans, Shie et al. [53] disclosed that repeated BTX-A injections significantly reduced the number of activated mast cells in the bladder. In another study, vascular endothelial growth factor (VEGF) expression and the apoptotic cell count were decreased in patients with IC/BPS after repeated BTX-A injections [54].

##### Increased Urothelial Permeability after Inflammation/Infection

Increased bladder permeability is believed to be a part of the underlying pathology of bladder hypersensitivity and hyperalgesia in IC/BPS patients or occur secondary to localized inflammation [45]. Many studies have pointed out that patients with IC/BPS, but not OAB, have a damaged or ulcerative, thin urothelium [55]. Liu et al. [56] confirmed results that identified mast cell infiltration in both the OAB and IC/BPS bladder wall, but showed reduced expression of the tight junction protein, zona occludens-1; E-cadherin was only detected in IC/BPS tissues. Glycosaminoglycan (GAG) replacement therapy with pentosane polysulfate (PPS) has been shown to improve bladder pain for some IC/BPS patients [57]. This may be due to the repair and recovery of a tight urothelial barrier maintained by GAG, the anti-inflammatory actions of PPS, and the inhibition of mast cell histamine release [58]. Although effects of BTX-A on urothelial barrier proteins have not been reported, it is reasonable to suggest that BTX-A injection may be beneficial for relieving bladder pain by inhibiting the localized inflammation of urothelium, reducing the numbers of activated mast cells [53], and blocking the subsequent release of inflammatory mediators, such as histamine, cytokines, and proteases, therefore desensitizing peripheral afferent nerve endings [59].

##### Amplified Sensory Symptoms after Inflammation

It has been observed that women with a history of recurrent urinary tract infection in childhood are more prone to be diagnosed with IC/BPS later in life [60]. This phenomenon may be due to a protective hypersensitive response during the remission process after a preceding bladder infection or inflammation. Basic studies have demonstrated that bladder insults in neonatal rats lead to a hypersensitive response to inflammation stimuli when tested in adults [61], and strengthens the spontaneous bladder distension-evoked activity of spinal visceral nociceptive neurons [62]. Altered spinal cord circuits regulate this situation because neonatal inflammation can prompt a downregulation of GABA (Aα-1) receptor microRNA and altered opioid peptide content in the dorsal horn [63,64].

### 2.3. BTX Effects in Bladder Urothelium and Lamina Propria

The bladder sensory system is complex and encompasses not only local afferent nerves, but also the bladder urothelium and lamina propria (LP), thus including the entire bladder mucosa. The urothelium was previously viewed as merely a passive blood-urine permeability barrier; however, it now apparently plays an active role in the bladder’s sensory system by having certain “neuronal-like properties” [65]. In vitro studies have shown that some neurotransmitters, including NO, ATP, Ach, and prostaglandins, are released from the urothelium after the application of chemical or physical stressors [66]. BTX-A is able to bind to the toxin’s receptor, SV2, within bladder urothelium and suppress hypotonic-evoked ATP release from rat urothelial cultures [24]. The LP lies between the urothelium and detrusor muscle, and contains mainly connective tissue, lymphatics, and abundant vasculatures [67]. The LP consists of afferent and efferent nerve endings, and acts as a “communication center” to integrate signals of the urothelium and local afferent nerve terminals [68]. Two specific kinds of cells, telocytes (Tc) and myofibroblasts (Myo), constitute a three-dimensional (3D) network structure in the LP that acts as a mass of stretch-receptors capable of perceiving physical and chemical stimuli and consequently behaving as a “functional syncytium” [69]. The Myo/Tc 3D network contributes to bladder compliance, avoids organ deformity and expresses muscarinic, vanilloid, and purinergic receptors that recognize signals from the urothelium and afferent nerve terminals to propagate information through this network to the bladder detrusor [69]. BTX-A was proposed to induce phenotypic changes in the Myo/Tc network, including the inhibition of expression of purinergic and SP receptors, and a reduction in the expression of contractile and gap junction proteins [70].

#### Nerve Sprouting and Exhaustion of BTX efficacy

The progressive loss of BTX-A efficacy can be seen during the treatment. When BTX-A was injected into a striated muscle, the efficacy persisted till antibodies against BTX-A were formed [71]. In the bladder, however, the phenomenon of losing BTX-A effectiveness may not work the same [68].

While BTX-A injection blocks nerve terminals, new nerve endings sprout to restore synaptic activity. Haferkamp et al. [72] biopsied the urothelium and LP of NDO patients, before and after the first BTX-A injection, and found axonal degeneration, nerve sprouting, and Schwann cell activation. In order to transduce signals correctly within the bladder sensory system, an appropriate distance between cells is necessary. Since sprouting is likely to be disorganized, the integration of signaling inside the LP system may be disturbed [71]. The excitation of new sprouting afferent nerve endings contributes to chronic neurogenic inflammation. Inflammation also activates the sensory nerve endings of the LP and causes the release of neuropeptides (SP, ATP, CGRP, neuropeptide Y) that mediate multidirectional interactions in Myo/Tc multicellular networks, and acts on endothelial, smooth muscle, and immune cells, and even back on nerve endings. These effects cause a positive feedback loop and turn into a vicious cycle [73]. The exhaustion of BTX efficacy is observed in NDO patients and may be due to the growth of afferent sprouts after repeated injections, which produce a chain reaction over time by maintaining and amplifying neurogenic inflammation [68].

### 2.4. BTX Effects in Central Nervous System

So far, the analgesic properties of BTX-A have been widely investigated in a variety of pain models. During investigations the following interesting observations were noted: (1) the effects induced by BTX-A administration are observed distantly from the site of injection; and (2) BTX-A affects not only the peripheral but also the central nervous system (CNS) [74]. In the experiments with radiolabeled BTX-A, the retrograde transport of BTX-A to the CNS has been recognized for decades [75]. However, the neurotoxin was thought to be possibly inactivated when reaching the CNS due to the slow rate of retrograde axonal transport [76]. Restani et al. observed that BTX-A was internalized by spinal cord motor neurons and underwent fast axonal retrograde transport by directly monitoring endocytosis and axonal transport of the neurotoxin [77].

The long-distance retrograde effects of BTX-A were thoroughly reported by Antonicci et al. [78]. They were the first to show that BTX-A applied in the peripheral nerves affected central circuits via retrograde transport and transcytosis [78]. The hypothesis of retrograde action of BTX-A in pain pathway is illustrated in Figure 2. In a rat bladder model, the concentrations of radiolabeled BTX-A increased over time in both L6-S1 dorsal root ganglia and L6-S1 spinal cord segments after injections it into the bladders of rats [79]. Because of its retrograde axonal transport to the CNS, the BTX-A neurotoxin may gain access to second-order neurons to affect these.

Interestingly, recent studies described how the unilateral injection of BTX-A can bilaterally reduce pain. These analgesic effects were shown in rat models of paclitaxel-induced peripheral neuropathy, carrageenan-induced hyperalgesia, and acidic saline-induced mirror pain [80,81,82]. Favre-Guilmard et al. [82] designed a carrageenan-induced pain model by subplantarly injecting BTX-A into experimental rats. Dramatic anti-hyperalgesia effects in uninjected contralateral hind paws of the rats were found in this study, which cannot be explained by the peripheral mechanism of BTX-A. These results suggested that BTX-A might have a central effect via the retrograde axonal transport system, which is also presumed to be the mechanism by which BTX-A acts to induce central neuropathic pain [83].

Intrathecal BTX-A administration by Coelho et al. [84] unlocks a brand new field of investigation into a deeper understanding of the actions of BTX-A [85]. This brilliant study, using animal models of severe bladder pain, described the administration of BTX-A via an alternative intrathecal route, which effectively functioned while undesirable side effects were avoided, including decreased detrusor pressure and increased post-void residual in bladder injections. This intrathecal route of administration was further investigated for intractable or refractory patterns of pain [6].

A recent breakthrough study using concurrent functional magnetic resonance imaging (fMRI) and urodynamic studies in female patients with multiple sclerosis and neurogenic overactive bladders reported that an intradetrusor injection of BTX-A increased the activity of most brain regions (cingulate body, prefrontal cortex, insula, and pontine micturition center) involved in the sensation and process of urinary urgency [86]. This was a pilot study to evaluate the possible effects of BTX-A at the level of the brain where sensory awareness is located. However, to date and to the best of our knowledge, no available evidence exists which directly demonstrates the effects of BTX-A on ascending bladder pain via a CNS system.

## 3. Conclusions

BTX-A is a promising option for treating bladder pain. Although the mechanisms involved are complicated, recent research efforts using a growing body of diverse expertise have been fruitful, especially in understanding the molecular architecture of the neurotoxin, as well as in the use of bioengineered animal models and in gaining electrophysiological-based insights. The analgesics effects of BTX-A are thought to be mainly mediated by muscle relaxation as well as the blockage of neurotransmitters and inflammatory substances. Recently, a hypothesis of BTX-A affecting the CNS via retrograde transportation to target neurotransmission in pain sensory circuits has been developed but is still very controversial when applied to humans [87]. Further research on the central action of BTX-A is important and will provide crucial information to better understand the pathophysiology of bladder pain. To date, BTX-A has only been approved by the U.S. Food and Drug Administration for NDO and OAB refractory to first-line therapy. This review comprehensively includes current molecular evidence of the effects of BTX-A on bladder pain. Further basic studies and clinical trials with a large number of patients are required in order to provide much more robust evidence-based support in using BTX-A to treat bladder pain.

## 4. Materials and Methods

This study is a literature review on the efficacy of BTX-A in bladder pain, focusing on molecular evidence and animal studies. A search for original and review articles was performed on PubMed, MEDLINE, Crossref, Embase, and Google Scholar databases using “botulinum toxin”, “molecular model”, “animal model”, and “bladder pain” as search terms. To expand the scope of the search, we included the following terms: “interstitial cystitis”, “hypersensitive bladder”, “neuropathic pain”, “pain management”, “neurotoxin”, “neurogenic bladder”, and “botulinum toxin injections”. This is a non-systemic review that was based on previously published articles. The search results were used to summarize current evidence for the possible molecular mechanism of action of BTX-A on bladder pain. All papers identified were English-language, full-text papers. We also checked the reference lists of selected articles to identify any papers with potentially missed data.

## Figures and Tables

**Figure 1 toxins-12-00098-f001:**
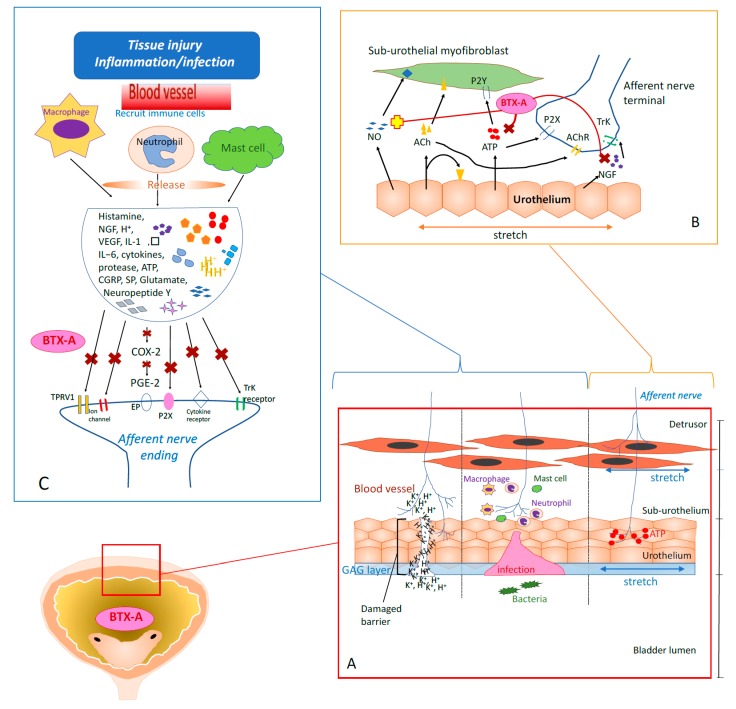
Mechanism of Intravesical BTX-A Effects on Peripheral Nervous System. (**A**) Afferent nerves innervate bladder sensation and carry information toward the central nervous system (CNS). Bladder-stretching is detected by the afferent nerve endings that extend into detrusor smooth muscles. The afferent nerve terminals extend into urothelium and sub-urothelial interstitium. These nerve terminals are sensitized when bacteria invade urothelium or high potassium ion penetrate after the urothelial barrier breaks down. (**B**) When bladder distention, the stretching urothelium releases neurotransmitters, including ATP, NGF, acetylcholine, and NO to activate afferent nerves. BTX-A blocks the afferent input by normalizing the balance of NO and ATP (blocks ATP and enhances NO). Besides, BTX-A dampens NGF, leads to attenuation of afferent excitability. (**C**) Immune cells including mast cells, macrophages and neutrophils were recruited by the cytokines released during bacterial infection or tissue damage. Histamine, interleukins, neuropeptides and more cytokines are subsequently released, which activates the bladder afferents.

**Figure 2 toxins-12-00098-f002:**
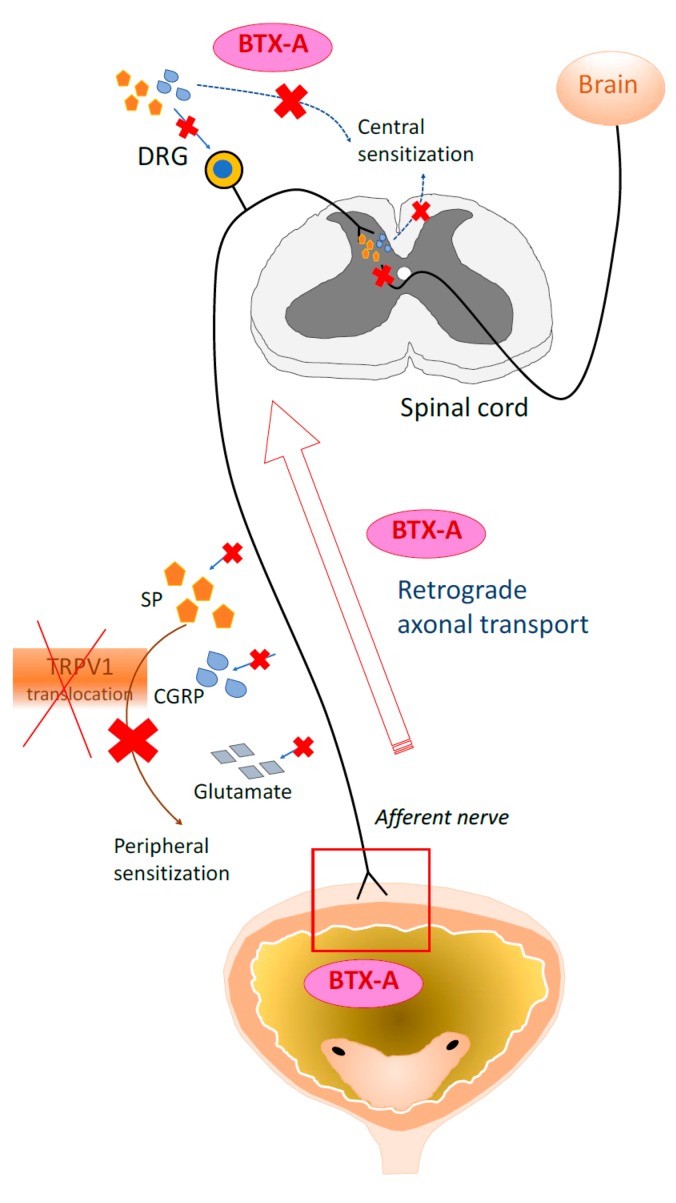
Illustration of Actions of BTX-A along the pain pathway. (1) BTX-A blocks the release of nociceptive neurotransmitters (SP, glutamate and CGRP) from afferent nerve endings and the translocation of TRPV receptor, leading to peripheral desensitization and results in attenuation of inflammation and pain. (2) Intravesical-injected BTX-A gains access to dorsal root ganglia (DRG) and spinal cord via retrograde axonal transport pathway. By blocking SP and CGRP there, BTX-A inhibits further central sensitization.

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
