# Peer review of "Effect of Botulinum Toxin A on Bladder Pain—Molecular Evidence and Animal Studies"

_toxins, 2020, doi:10.3390/toxins12020098_

Round 1

Reviewer 1 Report

The review titled ‘Effect of Botulinum Toxin A on Bladder Pain- Molecular Evidence and Animal Studies’ takes into consideration the possibility that the BTX-A could be used in some pain conditions localized in the lower urinary tracts and refractory to the common analgesic anti-inflammatory drugs available. The authors give a partial overview that needs to be integrated.

Indeed, the authors missed completely the data on BTX-A effects on the nerve terminals (and likely on the connective cells) located in the bladder lamina propria after acute and chronic administration of the toxin in NDO. In this regard, a review was recently published in Toxins. These data are important since, from one side, they reinforce the hypothesis of the authors that BTX-A acting  on the sensory terminals could play analgesic effects, from the other side, however, the chronic administration of the toxins could potentiate the pain symptomatology because of the sprouting of the excitatory nerve terminals and the neuroinflammation observed in the lamina propria of NDO patients.

I also would suggest intercalating the text with some schemes better showing the possible mechanisms of the analgesic activity of the toxin.

A deep revision of the text is necessary because there are several grammar and syntax mistakes all along the entire text. Further, the authors have a very peculiar (but wrong) way to use the particles instead of the substantives or vice-versa.  

Following are some examples that needs to be corrected:

Row 22: ‘Exposed ??? (Exposure) to the botulinum toxin could be fatal…’

Row 26: ‘Dykstra et al. was (were) the first…’

Rows 32-35: the two sentences are very difficult to be understood. Make them easier.

Rows 46-47: ‘There are…….have been identified-…’  too many verbs.

Row 48: ‘..the toxin was (is)...’

Row 77: ‘…(TRP) superfamily …….involves in many cellular function and they are highly…..’ (is involved in many cellular functions) and the cationic ion channels are highly….)

Row 80: ‘…involve..’ (are involved).

Row 89: ‘When bladder distension (relaxes), the ligand….’

Row 105: ‘….happened.’ remove the verb.

Row 108: ‘…serial..’????

Row 117: ‘Tons of studies had been proven..(have proven) that…’

Row 125: remove ‘which’

Row 164: ‘…that patients identified mast cell infiltration..??????’

Row 176: ‘…in women….., they are..’    Correct the sentence!

Row 179: ‘…a neonatal rat with bladder insult would lead to…’ Correct the subject of the sentence.

Row 186: ‘…something interesting were found that…(was found)’

Row 200: ‘…., recent studies had observed that…(have reported that..)’

Row 209: ‘… This genius study, (brilliant)..’.

Row 235: ‘…patient number were (are) inevitable for..’

Materials and Methods

It is necessary the authors extend their search to the papers suggested at the beginning.

Author Response

Dear Reviewer,

Thank you for your thoughtful and constructive feedback.

Point 1: The authors missed completely the data on BTX-A effects on the nerve terminals (and likely on the connective cells) located in the bladder lamina propria after acute and chronic administration of the toxin in NDO. In this regard, a review was recently published in Toxins.   

Response 1:

With respect to your wise point, we had already included this valuable review article as your suggestion and put the data in the results part, please see the paragraph titled: “2.3. BTX Effects in Bladder Urothelium and Lamina Propria”. The revised results were as follows, “Bladder urothelium is involved in the bladder’s sensory system by having certain “neuronal-like properties”. In vitro studies have shown that some neurotransmitters including NO, ATP, Ach, and prostaglandins, are released from the urothelium after the application of chemical or physical stressors [65]. BTX-A is able to bind to the toxin’s receptor, SV2, within bladder urothelium and suppress hypotonic-evoked ATP release from rat urothelial cultures [24]. The lamina propria (LP) lies between the suburothelium and muscularis propria, and contains mainly connective tissue, lymphatics, and abundant vasculatures [66]. The LP consists of afferent and efferent nerve endings, and acts as a “communication center” to integrate signals of the urothelium and local afferent nerve terminals [67]. Two specific kinds of cells exist: telocytes (Tc) and myocytes (Myo). These cells constitute a three-dimensional (3D) network structure in the LP and act as stretch-receptors. The Myo/Tc 3D network contributes to bladder compliance, avoids organ deformity and expresses muscarinic, vanilloid, and purinergic receptors that recognize signals from the urothelium and afferent nerve terminals to propagate information through the network to the bladder detrusor [68](Page 5, line 6-18)

    We also mentioned the sprouting of the excitatory nerve terminals and mechanism of the chronic neuroinflammation and possible following exhaustion of BTX-A efficacy. The additive paragraph is as follows, “While a BTX-A injection blocks nerve terminals, new nerve endings sprout to restore synaptic activity. Since sprouting is likely to be disorganized, the integration of signaling inside the LP system may be disturbed [69]. The excitation of new sprouting afferent nerve endings contributes to chronic neurogenic inflammation. Inflammation also activates the sensory nerve endings of the LP and causes the release of neuropeptides (SP, ATP, CGRP, neuropeptide Y), which mediate multidirectional interactions in Myo/Tc multicellular networks and acts on endothelial, smooth muscle, and immune cells, even acting back on nerve endings. These effects cause a positive feedback loop and turn into a vicious cycle [70]. The exhaustion of BTX efficacy is observed in NDO patients and may be due to the growth of afferent sprouts after repeated injections, which produce a vicious cycle over time by maintaining and amplifying neurogenic inflammation [67].” (Page 5, line 19-29)

Point 2: I also would suggest intercalating the text with some schemes better showing the possible mechanisms of the analgesic activity of the toxin.

Response 2:

    Thank you very much for the advice. We had added two schemes: Figure 1 showed “Mechanism of Intravesical BTX-A Effects on Peripheral Nervous System.” and figure 2 depicted “Illustration of Actions of BTX-A along the pain pathway.” In figure 1, we described the main mechanisms of peripheral sensitization in bladder, including bladder stretching, bacterial infection, and urothelial barrier break down. The analgesic effects of intravesical BTX-A were also demonstrated in the figure. Besides, the axonal retrograde transportation of BTX-A to attenuate central sensitization was illustrated in figure 2, which may help better understanding. Enclosed please find the attached figures.

Point 3: A deep revision of the text is necessary because there are several grammar and syntax mistakes all along the entire text.

Response 3:

      Thank you for your carefully reading. We are sorry for the gramma and syntax mistakes. We have checked and revised the entire text in detail and we also used a professional English editing service, which confirmed that the manuscript has been edited by native English speakers with a related biomedical background. Enclosed please find the attached certification for your reference.

      We really appreciate your patient reading and kind corrections. Following are the specific texts that we had corrected:

Row 22: ‘Exposed ??? (Exposure) to the botulinum toxin could be fatal…’

-->“Exposure to the botulinum toxin can be fatal after causing subsequent flaccid paralysis of the muscles and dysautonomia.” (Page 1, paragraph one, line 2)

Row 26: ‘Dykstra et al. was (were) the first…’ 

-->“Dykstra et al. were the first to use BTX-A in a urological application by injecting it into the urethral sphincter to treat the detrusor-sphincter dyssynergia in spinal cord injury patients.” (Page one, paragraph 1, line 6)

Rows 32-35: the two sentences are very difficult to be understood. Make them easier.

 --> We re-write the sentences as follows, “IC/BPS is a long-time challenge for urologists who treat its multifactorial conditions and accompanying pain. Recently, it was recognized that the disease not only has organ-specific syndromes, but also urogenital manifestations of regional or systemic abnormalities characterized by neuropathic pain.” (Page 1, line 2 from the bottom)

Rows 46-47: ‘There are…….have been identified-…’  too many verbs.

 --> The original content is: “There are two types of presynaptic cell membrane surface receptors of BTX-A have been identified — gangliosides and the synaptic vesicle associated protein-2 (SV2) family.” We corrected the sentence as below, “Two types of presynaptic cell membrane surface receptors for BTX-A have been identified—gangliosides and the synaptic vesicle associated protein-2 (SV2) family.” (Page 2, line 13-14)

Row 48: ‘..the toxin was (is)...’

--> We are sorry for the mistake and it was corrected as, “allowing the toxin is subsequently endocytosed into synaptic vesicles.” (Page 2, line 15)

Row 77: ‘…(TRP) superfamily …….involves in many cellular function and they are highly…..’ (is involved in many cellular functions) and the cationic ion channels are highly….)

--> Thank you again for the correction. The revised texts are as follows, “The transient receptor potential (TRP) superfamily of cationic ion channels is involved in many cellular functions and such channels are highly expressed in afferent neurons of the urinary bladder.” (Page 2, line 3 from the bottom)

Row 80: ‘…involve..’ (are involved).

--> We are sorry for the grammatical error. The sentence was corrected as below, “Members of the TRP channels superfamily include TRP vanilloid 1 (TRPV1), TRPV4, and TRP Ankyrin 1 (TRPA1), involved in the mechanosensory pathway of urothelial cells.” (Page 3, line 1)

Row 89: ‘When bladder distension (relaxes), the ligand….’

--> We paragraphed the sentence as “When the bladder is distended, ligand-gated ion channels P2X purinoceptors 3 (P2X3) receptors on nerve endings in the bladder urothelium are activated by released ATP and evoke a neural discharge.” (Page 3, line 10)

Row 105: ‘….happened.’ remove the verb.

--> Thank you for your kind suggestion. The verb was removed. Besides, we revised the sentence as below, “Under normal conditions, C-fibers are silent but become activated in several pathological conditions, such as the alteration of potassium channels.” (Page 3, line 26)

Row 108: ‘…serial..’????

--> We are sorry for the mistake and the word, “serial”, was deleted. The sentence had been revised to “Nerve growth factor (NGF) influences C-fibers hyperexcitability in studies done by Vizzard et al. [30] and Seki et al. [31].” (Page 3, line 30)

Row 117: ‘Tons of studies had been proven..(have proven) that…’

-->  “Numerous studies have shown that BTX-A blocks the release of nociceptive neurotransmitters from peripheral sensory nerves.” (Page 3, line 15 from the bottom)

Row 125: remove ‘which’

--> The word, “which”, was removed, and we revised the whole sentence like follows, “TRPV1 is a vanilloid receptor expressed in C-fibers that are involved in pain transmission after activation by heat, capsaicin, or resiniferatoxin.” (Page 3, line 7 from the bottom)

Row 164: ‘…that patients identified mast cell infiltration..??????’

--> We apologize for this inappropriate description. The sentence was revised as below, “Liu et al. [56] confirmed results that identified mast cell infiltration in both the OAB and IC/BPS bladder wall, but showed reduced expression of the tight junction proteins,…” (Page 4, line 17 from the bottom)

Row 176: ‘…in women….., they are..’    Correct the sentence!

--> Thank you for the kind reminder. We corrected the sentence as follows, “It has been observed that in women with a history of recurrent urinary tract infection in childhood are more prone to be diagnosed with IC/BPS later in life.”(Page 4, line 5 from the bottom)

Row 179: ‘…a neonatal rat with bladder insult would lead to…’ Correct the subject of the sentence.

--> The subject of the sentence was corrected. The corrected sentence is as below, “Basic studies have demonstrated that bladder insults in neonatal rats leads to a hypersensitive response to inflammation stimuli when tested as adults.” (Page 4, line 2 from the bottom)

Row 186: ‘…something interesting were found that…(was found)’

--> We are sorry for this grammatical error and it was corrected as follows, “During investigations the following interesting observations were noted” (Page 5, line 37)

Row 200: ‘…., recent studies had observed that…(have reported that..)’

--> This error had been corrected. We rewrote the sentence to “recent studies described how the unilateral injection of BTX-A can bilaterally reduce the pain.” (Page 6, line 1)

Row 209: ‘… This genius study, (brilliant)..’.

--> Thank you for the wise suggestion. We changed the adjective from “genius” to “brilliant”. (“This brilliant study, using animal models of severe bladder pain…” ) (Page 6, line 11-12)

Row 235: ‘…patient number were (are) inevitable for..’

--> Thank you for the correction. The sentence was revised as follows, “Further basic studies and clinical trials with a large number of patients are required in order to provide much more robust evidence-based support…” (Page 6, line 36)

Point 4: It is necessary the authors extend their search to the papers suggested at the beginning.

Response 4:

Thank you very much for your wise suggestion. We had extended our search as suggested and summarized the findings to an additional paragraph titled: “2.3. BTX Effects in Bladder Urothelium and Lamina Propria.” The content is as below:

“2.3. BTX Effects in Bladder Urothelium and Lamina Propria

Bladder urothelium is involved in the bladder’s sensory system by having certain “neuronal-like properties”. In vitro studies have shown that some neurotransmitters including NO, ATP, Ach, and prostaglandins, are released from the urothelium after the application of chemical or physical stressors [65]. BTX-A is able to bind to the toxin’s receptor, SV2, within bladder urothelium and suppress hypotonic-evoked ATP release from rat urothelial cultures [24]. The lamina propria (LP) lies between the suburothelium and muscularis propria, and contains mainly connective tissue, lymphatics, and abundant vasculatures [66]. The LP consists of afferent and efferent nerve endings, and acts as a “communication center” to integrate signals of the urothelium and local afferent nerve terminals [67]. Two specific kinds of cells exist: telocytes (Tc) and myocytes (Myo). These cells constitute a three-dimensional (3D) network structure in the LP and act as stretch-receptors. The Myo/Tc 3D network contributes to bladder compliance, avoids organ deformity and expresses muscarinic, vanilloid, and purinergic receptors that recognize signals from the urothelium and afferent nerve terminals to propagate information through the network to the bladder detrusor [68].

Nerve Sprouting and Exhaustion of BTX efficacy

While a BTX-A injection blocks nerve terminals, new nerve endings sprout to restore synaptic activity. Since sprouting is likely to be disorganized, the integration of signaling inside the LP system may be disturbed [69]. The excitation of new sprouting afferent nerve endings contributes to chronic neurogenic inflammation. Inflammation also activates the sensory nerve endings of the LP and causes the release of neuropeptides (SP, ATP, CGRP, neuropeptide Y), which mediate multidirectional interactions in Myo/Tc multicellular networks and acts on endothelial, smooth muscle, and immune cells, even acting back on nerve endings. These effects cause a positive feedback loop and turn into a vicious cycle [70]. The exhaustion of BTX efficacy is observed in NDO patients and may be due to the growth of afferent sprouts after repeated injections, which produce a vicious cycle over time by maintaining and amplifying neurogenic inflammation [67].”

Reviewer 2 Report

To the Authors;

This review has focused on bladder pain relief, including interstitial cystitis/bladder pain syndrome (IC/BPS) by Botulinum Toxin (BTX)-A introduction. In vitro mechanism, as well as in vivo mechanisms in animals, and human are described in BTX-A treated bladder pain relief. So many review articles regarding BTX-A-treated bladder pain relief have also been published year by year. The differences with the published past reviews are not clear, and the schematic figures suggesting the functional mechanism of BTX-A-induced bladder pain relief are not shown, too.  A Review describing BTX-A involved with urinary tract diseases should be arranged more in detail since abundant studies of bladder pain control using BTX-A have been accumulated so far. Therefore, the current manuscript is not suitable for publication in the journal, ‘Toxins”.

Author Response

Dear Reviewer, 

Thanks for your nice review. 

Point 1: The differences with the published past reviews are not clear.   

Response 1:

Thank you for the comment. Indeed, we divided the mechanisms of botulinum toxin A in managing pain into peripheral and central desensitization just as previously published articles. However, the previous review did not focus on the effects of BTX-A in bladder pain. Besides, we had reviewed the latest evidences and included the possible mechanism of exhaustion of BTX efficacy due to nerve sprouting. Furthermore, we also included the article published in 2019 by Khavari et al., which is the first study to evaluate the possible effects of intradetrusor injection of BTX-A at the human brain level. Therefore, our work is the most updated review.

Point 2: The schematic figures suggesting the functional mechanism of BTX-A-induced bladder pain relief are not shown.

Response 2:

We appreciate your advice and we had added two schemes: Figure 1 showed “Mechanism of Intravesical BTX-A Effects on Peripheral Nervous System.” and figure 2 depicted “Illustration of Actions of BTX-A along the pain pathway.” In figure 1, we described the main mechanisms of peripheral sensitization in bladder, including bladder stretching, bacterial infection, and urothelial barrier break down. The analgesic effects of intravesical BTX-A were also demonstrated in the figure. Besides, the axonal retrograde transportation of BTX-A to attenuate central sensitization was illustrated in figure 2. Please see the attached figures.

Reviewer 3 Report

The review in title Effect of Botulinum Toxin A on Bladder Pain—Molecular Evidence and Animal Studies aims to summarize current reports about Botulinum Toxin A for bladder pain. It is a nice review, however, current version is not completed version and not ready for review now, such as no key words. Meanwhile, extensive revision is needed before it could be accepted for publication:

More information about materials and methods needed be provided; A table is encouraged to list all key references; A mechanism graph is also highly encouraged.

Author Response

Dear Reviewer,

Thanks for your helpful review. 

Point 1: More information about materials and methods needed be provided

Response 1:

Thank you for the suggestion. More information about materials and methods were provided that we added the description of the complete searching terms and searching database. The paragraph was revised in the uploaded manuscript.

Point 2: A mechanism graph is also highly encouraged.

Response 2:

We agreed with the reviewer’s comment and two figures were added to our manuscript to describe the mechanism of the effects of BTX-A. Figure 1 showed “Mechanism of Intravesical BTX-A Effects on Peripheral Nervous System.” and figure 2 depicted “Illustration of Actions of BTX-A along the pain pathway.” In figure 1, we described the main mechanisms of peripheral sensitization in bladder, including bladder stretching, bacterial infection, and urothelial barrier break down. The analgesic effects of intravesical BTX-A were also demonstrated in the scheme. Besides, the axonal retrograde transportation of BTX-A to attenuate central sensitization was illustrated in figure 2. Enclosed please find the attached figures.

Round 2

Reviewer 1 Report

I appreciate the efforts the authors made in following the review ‘requests. However, the manuscript still needs a revision as follow:

1)            The new two added paragraphs need to be deeply re-written to correct some mistakes and to make them conform to the text.

2)            There are still several grammar and syntax mistakes (some of which are listed below) and another revision by a native English person is necessary

Abstract

Row 8: remove …’a’ before refractory.

Introduction

Row 23: remove ‘which’

Rows 24-25. The sentence: ‘Exposure to the botulin toxin…’ is incorrected.

Row 28: ‘..relevant form.’ What for? Complete the sentence to give the meaning of the phrases.

Row 41: add: cellular after ..’molecular..’

Results

Rows 51-52: ‘ …allowing the toxin is  subsequently endocytosed into synaptic vesicles’ change as follow:

 ‘allowing the toxin to be endocytosed into …’

Rows 151-152: ‘.., animal models of cystitis have employed…’ Attention: the animal models do not employed something. Write:.. in animal model have been employed.

Rows 165-166: ‘..have a…diminished urothelium.’???

Rows 192-194: ‘The lamina propria….’ Attention: the lamina propria correspond to the layer between the urothelium and the detrusor, it contains the suburothelium and the submucosa.

Row 196: Attention ‘myofibroblasts’ NO myocytes (the latter are smooth muscle cells)

Author Response

Dear Reviewer, 

Thank you for your delicate and careful review. 

Point 1: The new two added paragraphs need to be deeply re-written to correct some mistakes and to make them conform to the text.   

Response 1:

Thank you for your advice, we had already added some contents and corrected the mistakes as you pointed out. Please see the revised paragraph titled: “2.3. BTX Effects in Bladder Urothelium and Lamina Propria”. The revised sentences were underlined for reference.

   2.3. BTX Effects in Bladder Urothelium and Lamina Propria

The bladder sensory system is complex and encompasses not only local afferent nerves, but also the bladder urothelium and lamina propria (LP), thus including the entire bladder mucosa. The urothelium was previously viewed as merely a passive blood-urine permeability barrier; however, it now apparently plays an active role in the bladder’s sensory system by having certain “neuronal-like properties” [65]. In vitro studies have shown that some neurotransmitters, including NO, ATP, Ach, and prostaglandins, are released from the urothelium after the application of chemical or physical stressors [66]. BTX-A is able to bind to the toxin’s receptor, SV2, within bladder urothelium and suppress hypotonic-evoked ATP release from rat urothelial cultures [24]. The LP lies between the urothelium and detrusor muscle, and contains mainly connective tissue, lymphatics, and abundant vasculatures [67]. The LP consists of afferent and efferent nerve endings, and acts as a “communication center” to integrate signals of the urothelium and local afferent nerve terminals [68]. Two specific kinds of cells, telocytes (Tc) and myofibroblasts (Myo), constitute a three-dimensional (3D) network structure in the LP that acts as a mass of stretch-receptors capable of perceiving physical and chemical stimuli and consequently behaving as a “functional syncytium” [69]. The Myo/Tc 3D network contributes to bladder compliance, avoids organ deformity and expresses muscarinic, vanilloid, and purinergic receptors that recognize signals from the urothelium and afferent nerve terminals to propagate information through this network to the bladder detrusor [69]. BTX-A was proposed to induce phenotypic changes in the Myo/Tc network, including the inhibition of expression of purinergic and SP receptors, and a reduction in the expression of contractile and gap junction proteins [70].

Nerve Sprouting and Exhaustion of BTX efficacy

The progressive loss of BTX-A efficacy can be seen during the treatment. When BTX-A was injected into a striated muscle, the efficacy persisted till antibodies against BTX-A were formed [71]. In the bladder, however, the phenomenon of losing BTX-A effectiveness may not work the same [68]. While a BTX-A injection blocks nerve terminals, new nerve endings sprout to restore synaptic activity. Haferkamp et al. [72] biopsied of the urothelium and LP of NDO patients, before and after the first BTX-A injection, and found axonal degeneration, nerve sprouting, and Schwann cell activation. In order to transduce signals correctly within the bladder sensory system, an appropriate distance between cells is necessary. Since sprouting is likely to be disorganized, the integration of signaling inside the LP system may be disturbed [71]. The excitation of new sprouting afferent nerve endings contributes to chronic neurogenic inflammation. Inflammation also activates the sensory nerve endings of the LP and causes the release of neuropeptides (SP, ATP, CGRP, neuropeptide Y) that mediate multidirectional interactions in Myo/Tc multicellular networks, and acts on endothelial, smooth muscle, and immune cells, and even back on nerve endings. These effects cause a positive feedback loop and turn into a vicious cycle [73]. The exhaustion of BTX efficacy is observed in NDO patients and may be due to the growth of afferent sprouts after repeated injections, which produce a chain reaction over time by maintaining and amplifying neurogenic inflammation [68].

Point 2: There are still several grammar and syntax mistakes (some of which are listed below) and another revision by a native English person is necessary.

Response 2:

         Thank you very much for the kind corrections. We had corrected the mistakes point-by-point and we also sent for a second-round professional English editing by a native English speaker with a related biomedical expertise.

Following are the specific texts that we had corrected:

Abstract

Row 8: remove …’a’ before refractory.

--> We had deleted the ‘a’. The revised sentence is as follows, “Although the only approved indications for BTX-A in the bladder are neurogenic detrusor overactivity and refractory overactive bladder,……”

Introduction

Row 23: remove ‘which’

--> We had deleted the ‘which’ and revised sentence as below, “Botulinum toxin, one of the most powerful neurotoxins in nature, is produced by the anaerobic, Gram-positive organism,……”

Rows 24-25. The sentence: ‘Exposure to the botulin toxin…’ is incorrected.

--> Thank you for the correction. We had corrected the sentence as follows, Exposure to the botulinum toxin can be fatal since this can lead to flaccid paralysis of the muscles, dysautonomia, and subsequent respiratory failure.

Row 28: ‘..relevant form.’ What for? Complete the sentence to give the meaning of the phrases.

--> Thank you for the advice. We had revised the sentence as below, “botulinum toxin A (BTX-A) shows the longest duration of activity in blocking transmission at the neuromuscular junctions, making it the most popular form for clinical use.

Row 41: add: cellular after ..’molecular..’

--> Thank you for the suggestion, and we had added the word in the sentence as follows, “Here we reviewed current molecular and cellular evidence

Results

Rows 51-52: ‘ …allowing the toxin is subsequently endocytosed into synaptic vesicles’ change as follow: ‘allowing the toxin to be endocytosed into …’

--> Thank you for your correction. We had changed as follows, “BTX-A binds to nerve terminals because of the high affinity of its heavy chain for SV2 allowing the toxin to be endocytosed into synaptic vesicles”

Rows 151-152: ‘.., animal models of cystitis have employed…’ Attention: the animal models do not employed something. Write:.. in animal model have been employed.

--> Thank you very much for the explanation. We had revised the sentence as below, “To investigate the inflammatory-mediated pathophysiology of bladder pain, a range of irritants or immune stimulants, including CYP, lipopolysaccharide, acetic acid, acrolein, and protamine sulfate, have been administered in animal models.

Rows 165-166: ‘..have a…diminished urothelium.’???

--> We are sorry for our ambiguous description. We had modified the wording as follows, “Many studies have pointed out that patients with IC/BPS, but not OAB, have a damaged or ulcerative, thin urothelium.

Rows 192-194: ‘The lamina propria….’ Attention: the lamina propria correspond to the layer between the urothelium and the detrusor, it contains the suburothelium and the submucosa.

--> Thank you very much for the correction. We are sorry for the mistake. We had corrected the text as below, “The LP lies between the urothelium and detrusor muscle, and contains mainly connective tissue, lymphatics, and abundant vasculatures.

Row 196: Attention ‘myofibroblasts’ NO myocytes (the latter are smooth muscle cells)

--> Thank you for the correction. We apologize for the error. We had corrected as follows, “Two specific kinds of cells, telocytes (Tc) and myofibroblasts (Myo), constitute a three-dimensional (3D) network structure in the LP

Again, thank you very much for the generous comments on the manuscript. We appreciate your time and energy in helping us improve the quality of our paper.

Reviewer 2 Report

To Authors,

Authors revised all of the issues of their manuscript with illustrated schemes, and the manuscript is suitable for the publication of the Toxins.

Author Response

Dear Reviewer, 

Thank you for the comment.

Reviewer 3 Report

The authors made nice revision. No more comments. 

Author Response

Dear Reviewer, 

Thank you very much for the review.